# OFFLINE MODEL-BASED SKILL STITCHING

## ABSTRACT

We study building agents capable of solving long-horizon tasks using offline model-based reinforcement learning (RL). Existing RL methods effectively learn individual skills. However, seamlessly combining these skills to tackle long-horizon tasks presents a significant challenge, as the termination state of one skill may be unsuitable for initiating the next skill, leading to cumulative distribution shifts. Previous works have studied skill stitching through online RL, which is time-consuming and raises safety concerns when learning in the real world. In this work, we propose a fully offline approach to learn skill stitching. Given that the aggregated datasets from all skills provide diverse and exploratory data, which likely includes the necessary transitions for stitching skills, we train a dynamics model designed to generalize across skills to facilitate this process. Our method employs model predictive control (MPC) to stitch adjacent skills, using an ensemble of offline dynamics models and value functions. To mitigate overestimation issues inherent in models learned offline, we introduce a conservative approach that penalizes the uncertainty in model and value predictions. Our experimental results across various benchmarks validate the effectiveness of our approach in comparison to baseline methods under offline settings.

## 1 INTRODUCTION

In the context of sequential decision making, an essential capability for intelligent agents is to solve long-horizon tasks composed of multiple stages. A long-horizon task can be viewed as an ordered combination of several "sub-tasks" or "skills". For example, consider a scenario where an intelligent robot has independently learned two skills: boiling water (skill-1) and pouring water into a glass (skill-2). To complete the task of "giving me a cup of boiled water", directly executing these skills in sequence could lead to significant failure, as the robot may struggle to pour water correctly after executing the boiling water skill. The termination state of skill-1 may be unsuitable to execute skill-2, especially when there are large discrepancies in the data used to train the different skills. Therefore, it is crucial to introduce a policy for stitching adjacent skills together when dealing with long-horizon tasks.

Reinforcement learning (RL) has emerged as a dominant paradigm for solving sequential decision making tasks, demonstrating remarkable capabilities in learning diverse real-world skills (Margolis & Agrawal, 2023; Yuan et al., 2023). While RL achieves excels in simpler tasks with relatively fixed goals, managing long-horizon tasks that require multiple sequential skills remains an unsolved challenge. Long-horizon tasks involve more transitions and require higher-level multi-step reasoning, making it difficult for RL to learn efficiently with a single policy. Additionally, if we learn each skill individually using RL, the gap between the transitions observed for different skills can be significantly large, complicating the direct chaining of these skills. To address these challenges, we propose to learn skill stitching to facilitate the integration of RL skills.

Previous studies have explored learning and stitching skills using online RL (Konidaris & Barto, 2009; Bagaria & Konidaris, 2019; Gu et al., 2023), which can be time-consuming and sample-inefficient. Other works (Lee et al., 2019; 2021; Kang & Oh, 2022; Chen et al., 2023) have managed to pre-train skills from offline datasets but still rely on online RL for stitching. This online learning stage introduces inefficiencies and safety concerns, especially in real-world settings. For example,

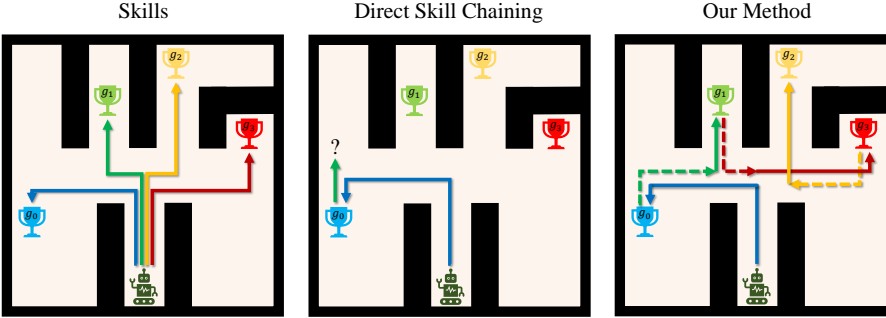

Figure 1: An example of the maze environment to illustrate skill stitching. Suppose we have four skills, denoted by the goals $\{g_i\}_{i=0}^3$. Consider a long-horizon task $g_0 \to g_1 \to g_3 \to g_2$ starting from the initial location at the bottom center of the map. Each of the three figures illustrates: (*left*) trajectories of these skills, distinguished by different colors; (*mid*) a case where skills are executed directly without stitching, i.e., skill executes one after another; (*right*) our approach where the agent successfully accomplishes the whole long-horizon task through skill stitching. In (*mid*), the agent struggles to accomplish the second skill, as the termination state of the first skill ($g_0$) is unobserved during the training of the second skill. In (*right*), we show the entire trajectory with skills distinguished by different colors, where dashed lines denote the stitching process and solid lines represent skill execution.

robots are often unsuitable for online RL due to inherent risks and safety issues. To address these concerns, can we learn both skills and stitching policies from offline data?

In this paper, we propose a fully offline method to tackle long-horizon tasks through skill stitching. We assume access to various skills and their associated datasets, recognizing that high-quality datasets for long-horizon tasks can be challenging to obtain. A proper combination of these skills can effectively accomplish a range of long-horizon tasks. Our primary motivation is that the aggregated datasets of all skills – whether from training datasets or RL replay buffers – offer rich coverage of environment transitions, which can be leveraged to learn skill stitching. We propose to train a dynamics model on these aggregated datasets, capturing environment transitions independent of specific skills, thereby demonstrating promising generalization for predicting transitions in states not observed by a specific skill. Additionally, we train each individual skill using offline RL with the skill datasets. With the offline-trained skills and dynamics models, we can perform model-based planning to facilitate the stitching process, which is integrated between two adjacent skills during the execution of long-horizon tasks. The planning process of model-based stitching is guided by the value function of the next skill, encouraging the agent to visit states that are suitable for executing that skill. For example, in a maze environment, as illustrated in Figure 1, the right figure demonstrates how the agent can accomplish the long-horizon task through skill stitching, while the middle figure shows that chaining skills directly without stitching leads to failure.

Since dynamics models and skill value functions are trained offline, prediction errors can increase when encountering states outside the training distribution. This results in significant cumulative error in model-based planning, as the planning algorithm utilizes the overestimated values of inaccurately predicted states to maximize its objective. To address this issue, we introduce a conservative approach for training offline models and conducting model-based planning. We train an ensemble of value functions for each skill and an ensemble of dynamics models. During the stitching process, we incorporate the variance of both model predictions and value predictions into the optimization objective, discouraging the agent from visiting risky out-of-distribution states.

In our experiments, we evaluate our method across three distinct benchmarks, including a maze environment and two robotic manipulation benchmarks – Kitchen (Fu et al., 2020) and Robosuite (Zhu et al., 2020). The results demonstrate that our method effectively manages diverse combinations of skills in each environment, outperforming various baselines in fully offline settings.

To summarize, our main contributions are as follows:

- We introduce the problem of offline skill stitching, tackling the challenges of solving long-horizon tasks with skill-based reinforcement learning.

- Our technical contributions – training dynamics models on aggregated skill datasets, stitching skills through model-based planning, and the conservative optimization objectives – provide an effective framework for fully offline skill stitching.

- Our method demonstrates improved performance over various baselines, highlighting the promising potential of offline model-based algorithms in the realm of skill stitching and long-horizon task execution.

## 2 Preliminaries

### 2.1 Problem Formulation

Consider $n$ different sub-tasks in a certain environment. Following prior work on offline reinforcement learning (RL), we assume access to datasets for training all the skills $\{\mathcal{D}_i\}_{i=1}^n$, where each $\mathcal{D}_i$ contains the trajectories of accomplishing a sub-task $i$. The task is represented as a Markov decision process (MDP) $\mathcal{M}$ defined by a tuple $\langle \mathcal{S}, \mathcal{A}, \mathcal{P}, r, \gamma, \rho_0 \rangle$, where $\mathcal{S}$, $\mathcal{A}$, $\gamma$, and $\rho_0$ represent the state space, action space, discount factor, and the distribution of initial states, respectively. $\mathcal{P} : \mathcal{S} \times \mathcal{A} \times \mathcal{S} \rightarrow [0,1]$ denotes the transition function, and $r : \mathcal{S} \times \mathcal{A} \rightarrow \mathbb{R}$ is the reward function. We denote the skill set as $\{\pi_i\}_{i=1}^n$, which are the policies to accomplish each single sub-task $i$. Formally, the objective is to learn the joint policy $\Pi$ so as to maximize the expected return in long-horizon tasks:

$$R = \mathbb{E}_{a_t \sim \Pi(\cdot|s_t), s_{t+1} \sim \mathcal{P}(\cdot|s_t, a_t)} \left[ \sum_t \left( \gamma^t r(s_t, a_t) \right) \right], \qquad (1)$$

where the joint policy $\Pi$ is relevant to both the skills $\{\pi_i\}_{i=1}^n$ and the stitching processes between skills.

Our goal is to leverage both the offline datasets $\{\mathcal{D}_i\}_{i=1}^n$ and the skill set $\{\pi_i\}_{i=1}^n$ to tackle any long-horizon task that necessitates sequential execution of a subset of skills. Thus, we aim to design an algorithm that can leverage offline datasets for stitching a sequence of skills $(\pi_{i_1}, ..., \pi_{i_n})$ into a joint policy $\Pi = (\pi_{i_1}, \pi_{i_1,i_2}^s, \pi_{i_2}, ..., \pi_{i_{n-1},i_n}^s, \pi_{i_n})$, so as to tackle the long-horizon task. Here, each $\pi_{i_k}$ is a skill required to solve the task that is learned from the corresponding skill dataset; each $\pi_{i_k,i_{k+1}}^s$ denotes the stitching process to chain the adjacent skills $i_k$ and $i_{k+1}$.

### 2.2 Offline Model-Based Reinforcement Learning

Offline reinforcement learning (Levine et al., 2020) is a branch of RL algorithms where agents have no access to interacting with the environment to collect trajectories, experiences and data. Instead, a static dataset of transitions $\mathcal{D} = \{(s_t, a_t, r_t, s_{t+1})\}$ is provided for training. The core issue is to get a sufficient understanding of the underlying MDP through limited data, so as to learn a good policy. Typical algorithms in offline RL include batch-constrained Q-learning (BCQ) (Fujimoto et al., 2019), conservative Q-learning (CQL) (Kumar et al., 2020), implicit Q-learning (IQL) (Kostrikov et al., 2022), etc.

In offline RL, a main problem is the overestimation in value of the out-of-distribution states/actions (unseen in the dataset) due to the distributional shift between the dataset and the learned policy (Kumar et al., 2020). To tackle this problem, CQL learns a conservative Q-function by producing a lower bound on the value of the current policy, while IQL introduces expectile regression to lower the estimation. Besides, value ensemble (An et al., 2021) is also widely used to reduce the overestimation, based on the extent of uncertainty during estimation.

Model-based RL (Janner et al., 2019) is put forward with the idea of modeling the transition (dynamics) of the environment directly. Specifically, the sampled experiences or the collected datasets are used to fit a dynamics model approximately, which can become a substitution for the real environment to generate "virtual" transition data. These synthesized data can reduce the need of real data from the environment, which is much more sample-efficient than model-free methods.

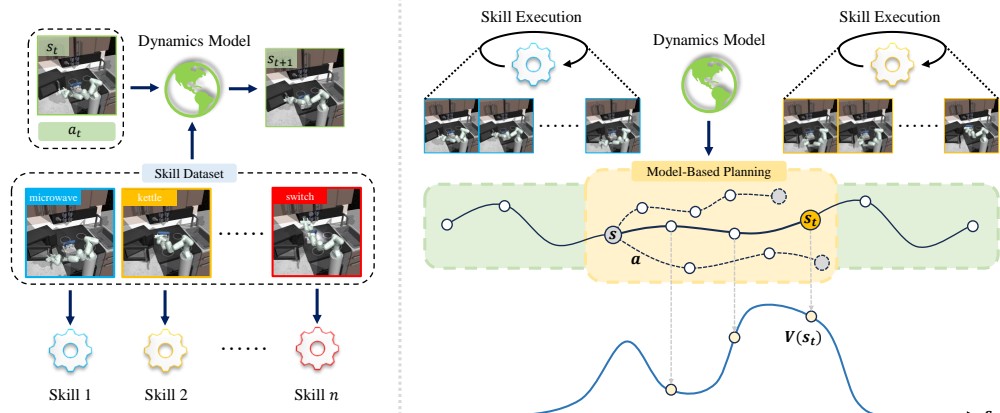

Figure 2: An overview of our offline skill stitching method. **Offline training (left):** We acquire skills through offline RL using skill datasets, which are used to compose long-horizon tasks. An ensemble of dynamics models is trained from the aggregated datasets to model the transitions in the environment, which is used to perform model-based planning for skill stitching. **Stitching (right):** We insert a stitching process between the executions of two adjacent skills to enhance long-horizon task execution. Our stitching method employs model-based planning, where the optimization objective is guided by the skill value functions and regularized with uncertainties in model predictions, leading the agent to new states that are suitable for executing the subsequent skill.

Combined with the setting of offline RL, the dynamics model is directly trained from the given dataset in model-based offline RL. In this case, the dynamics model plays an important role in the approximate substitution of the environment, since there is no access to interacting with the real environment in offline RL.

## 3 METHOD

Our method can be divided into the offline training phase and the test phase, as shown in Figure 2. During the training phase, we train policies of all skills using the skill datasets, and the dynamics model is trained using the aggregated datasets of all skills. For the test phase, we tackle long-horizon tasks through model-based skill stitching, enabling chaining adjacent skills effectively.

### 3.1 LEARNING SKILLS AND DYNAMICS MODELS

To acquire all the skills, we use the skill datasets $\{\mathcal{D}_1, \mathcal{D}_2, ..., \mathcal{D}_n\}$, where each dataset $\mathcal{D}_i$ contains either expert or sub-optimal trajectories for skill $i$. We train $n$ policies $\left\{\pi_1^{\theta_1}, \pi_2^{\theta_2}, ..., \pi_n^{\theta_n}\right\}$ correspondingly using IQL (Kostrikov et al., 2022), with each policy $\pi_i^{\theta_i}$, parameterized by $\theta_i$, manages skill $i$. To acquire a generalizable dynamics model that is independent of specific skills, we aggregate all the skill datasets $\bigcup_{i=1}^{n} \mathcal{D}_i$ and train the dynamics model $T_\phi(\cdot \mid s_t, a_t)$ parameterized by $\phi$ to minimize MSE loss on next state prediction. This aggregated dataset offers enhanced coverage of transitions within the environment, enabling the learned dynamics model to accurately predict the transitions needed for skill stitching. In the subsequent test phase, this dynamics model will be used for model-based skill stitching.

### 3.2 MODEL-BASED PLANNING FOR SKILL STITCHING

During the test phase, we adopt model predictive control (MPC) to plan an action sequence to stitch the adjacent skills. Specifically, we use random shooting method (Press, 2007; Bonnans, 2013) for the environments with discrete action spaces, and cross entropy method (CEM) (Rubinstein, 1999; De Boer et al., 2005) is adopted for those with continuous action spaces.

---

**Algorithm 1** Model-Based Planning for Stitching Skills

---

// **Training Phase:**

Given the datasets of all skills $\{\mathcal{D}_1, \mathcal{D}_2, ..., \mathcal{D}_n\}$

Train policies $\left\{\pi_1^{\theta_1}, \pi_2^{\theta_2}, ..., \pi_n^{\theta_n}\right\}$ and corresponding value functions $\left\{V_1^{\theta_1}, V_2^{\theta_2}, ..., V_n^{\theta_n}\right\}$ using IQL

Train the dynamics model $T_\phi(\cdot \mid s_t, a_t)$ using all the datasets $\bigcup_{i=1}^{n} \mathcal{D}_i$

// **Test Phase:**

Given a long-horizon task, denoted as a skill sequence

Reset the environment and get initial state $s_0$

Current state $s_c \leftarrow s_0$

**for** each skill $i$ in the long-horizon task **do**

    // Stitching Phase: add connections from skill $i-1$ to skill $i$

    The value of current state $v_{\text{new}} \leftarrow V_i^{\theta_i}(s_c)$

    **repeat**

        $v_{\text{old}} \leftarrow v_{\text{new}}$

        Use MPC to plan the next action $a_t$

        Execute action $a_t$ and get next state $s_{t+1}$

        The value of current state $v_{\text{new}} \leftarrow V_i^{\theta_i}(s_{t+1})$

    **until** $v_{\text{new}} < v_{\text{old}}$ or reaching maximum steps of stitching

    // Execution Phase: accomplish sub-task $i$

    **repeat**

        Execute policy $\pi_i^{\theta_i}$ in the environment

    **until** success or reaching maximum steps of skill execution

**end for**

---

We evaluate the quality of the new state by the value function of the IQL policy of the next skill, meaning that a state with a higher value (from the perspective of the next skill) tends to be a better state for executing the next skill. The planning stage loops as long as the predicted value of the new state is higher than the previous one, forming a monotonically increasing sequence. When the value stops increasing or the number of stitching steps exceeds the maximum value, stitching is terminated.

To reduce the problem of overestimation in value prediction, we adopt the trick of ensemble and train multiple models of the value function of IQL and the dynamics models. Since we predict the new state given sampled actions via dynamics models and predict the value of the new state using the value functions of IQL, the inaccuracy comes from both the value functions and the dynamics models. In practice, we calculate the ensemble value $v_{\text{ensemble}}$ following Equation 2:

$$v_{\text{ensemble}}(s_{t+1}) = \mathbb{E}\left[v(s_{t+1})\right] - \alpha \text{Var}(v(s_{t+1})) - \beta \text{Var}(T_\phi(s_{t+1} \mid s_t, a_t)), \quad (2)$$

where $v(\cdot)$ denotes the value of a state predicted by the value function of the IQL policy, and $s_{t+1} \sim T_\phi(\cdot \mid s_t, a_t)$ represents the predicted next state via the dynamics model. We use the average of multiple models to fit the expectation approximately, and use their standard deviation to approximate the variance.

## 4 EXPERIMENTS

### 4.1 EXPERIMENTAL SETUP

#### 4.1.1 ENVIRONMENTS AND TASKS

We conduct experiments on three different benchmarks, varying from the basic maze runners to complicated robotics simulated environment suites:

**MazeRunner.** This environment is redefined and modified from AMAGO (Grigsby et al., 2024). An agent spawns with a fixed location in a maze, whose map is a fixed $9 \times 9$ discrete gridworld.

The agent needs to navigate in the maze, and move to the given target location. We set four targets in this environment, and correspondingly, the datasets for training skill policies only contain four kinds of trajectories - from the initial spawn point to one of the goals. The dataset for training the dynamics model contains all kinds of transitions in the environment, so as to make the dynamics model unbiased. A long-horizon task is defined as continuous navigation among multiple goals, e.g., from the spawn point to goal-1, then from goal-1 to goal-2 directly.

**Kitchen.** A Franka robot works in a kitchen-like environment to accomplish simulated tasks in the kitchen (Fu et al., 2020). This benchmark provides 7 tasks (skills), including `bottom burner`, `top burner`, `light switch`, `slide cabinet`, `hinge cabinet`, `microwave`, and `kettle`. In this benchmark, the long-horizon tasks are defined as the combinations of multiple skills, e.g. `microwave` → `kettle` (open the microwave first and move the kettle afterwards).

**Robosuite.** This benchmark (Zhu et al., 2020) provides a variety of manipulation tasks in robotics, including `Door` (open the door by twisting the doorknob) and `PickPlaceCan` (pick a can from one side and place it into the box on the other side). In this paper, we adopt the skills and follow the action spaces of MAPLE (Nasiriany et al., 2022). We consider two kinds of combinations:

- `Door` → `CloseDoor` (open the door first and close it afterwards);
- `PickPlaceCan` → `PickPlaceCan` (pick and place the can twice).

Note that all the long-horizon tasks must be accomplished in the given order, e.g., for a long task `microwave` → `kettle`, the agent must open the microwave first and move the kettle afterwards. If the agent successfully moves the kettle without opening the microwave, it cannot receive the score (reward).

### 4.1.2 BASELINES

As discussed before, the most natural idea can be directly train a policy from the dataset of long-horizon tasks, while in most cases such long datasets are unavailable. Additionally, the policies directly derived from long-horizon datasets can only generally accomplish the tasks with the same order of skills. We provide experiments of this as an ablation study in Section 4.3.1.

We denote our method as **MB-Stitching**, since we add model-based planning as the stitching part between two adjacent skills with IQL policies. Specifically, we utilize random shooting method (Press, 2007; Bonnans, 2013) as MPC planning in MazeRunner, by generating random actions from a uniform distribution. For the benchmarks with continuous states and actions, we adopt cross entropy method (CEM) (Rubinstein, 1999; De Boer et al., 2005), sampling the actions from a Gaussian distribution and optimize the mean and standard until convergence. For comparison, we consider the following baselines in the domain of fully offline RL:

**No-Stitching.** We adopt implicit Q-learning (IQL) itself as the most basic baseline. The policies of skills are executed, without any stitching. In implementation, we use one multi-task policy (i.e., the goal-conditioned policy) instead of multiple single-task policies for skill execution in MazeRunner and Kitchen. In the Maze environment, a single policy can naturally handle four different skills, since the goal information is naturally contained within the observations of the Maze environment. In terms of the Kitchen environment, following the tradition of vector-based RL, we use a 7-dimensional one-hot vector (since there are 7 tasks in Kitchen) to denote which skill is being executed, and the one-hot vector is concatenated with the observation vector before taken by the policy network as input.

**Random-Stitching.** Randomly generated actions are used for the stitching part. The terminate function still holds the same as **MB-Stitching**, i.e., either the (predicted) value of the new state stops to increase monotonically or the number of stitching steps exceeds a maximum threshold.

**MF-Stitching.** This is a model-free baseline in the domain of offline RL. An extra policy is trained to serve as the stitching part, which is executed between two adjacent skills. This extra policy is offline trained from a "mixed" dataset, in which the second half of the trajectories of the previous skill and the first half of the trajectories of the next skill are directly mixed together.

Table 1: Average scores of skill stitching on MazeRunner. We evaluate our method as well as the baselines on stitching **all possible permutations** of two, three, and four skills, with each tested for 100 times for an average. For example, the results of "three goals" is the average of $A_4^3 = 24$ combinations of skills, with each combination tested 100 times for calculating the average. The score is equal to the number of accomplished skills divided by the total number of skills, i.e., the normalized score. We test on three different seeds and report the mean and variance.

| Tasks | Methods | | | |
|---|---|---|---|---|
| | No-Stitch | Random-Stitch | MF-Stitch | MB-Stitch (Ours) |
| 2 goals | 0.584±0.000 | 0.624±0.001 | 0.637±0.001 | **0.986±0.000** |
| 3 goals | 0.389±0.000 | 0.425±0.000 | 0.433±0.001 | **0.968±0.001** |
| 4 goals | 0.292±0.000 | 0.319±0.001 | 0.329±0.001 | **0.946±0.001** |

Table 2: Average scores of skill stitching on Kitchen. We evaluate our method as well as the baselines on stitching **all possible permutations** of two, three, and four skills, with each tested for 100 times for an average. For example, the results of "four skills" is the average of $A_7^4 = 840$ combinations of skills, with each combination tested 100 times for calculating the average. The score is equal to the number of accomplished skills divided by the total number of skills, i.e., the normalized score. We test on three different seeds and report the mean and variance.

| Tasks | Methods | | | |
|---|---|---|---|---|
| | No-Stitch | Random-Stitch | MF-Stitch | MB-Stitch (Ours) |
| 2 skills | 0.224±0.001 | 0.123±0.003 | 0.227±0.001 | **0.259±0.001** |
| 3 skills | 0.166±0.000 | 0.092±0.000 | 0.165±0.000 | **0.187±0.000** |
| 4 skills | 0.125±0.000 | 0.074±0.000 | 0.128±0.000 | **0.147±0.000** |

## 4.2 RESULTS AND ANALYSES

The test results of baselines and our methods on MazeRunner are listed in Table 1. **No-Stitching** holds a constant average score on 2, 3, and 4 goals, since it never accomplishes the third skill. **Random-Stitching** and **MF-Stitching** gains only little increase from 2 goals to 4 goals, which also implies that the baselines can hardly tackle the remaining skills after the first one. By contrast, the average score of **MB-Stitching** (our method) is extremely increasing as long-horizon tasks with more goals are tested, indicating that our stitching method is significantly effective.

The results on Kitchen are shown in Table 2. Since the datasets of kitchen are relatively low in both quantity and quality, the **No-Stitching** baseline are relatively weak even to accomplish the first skill. As shown in Table 8 in Appendix C, the low average score here results from the fact that a large quantity of combinations of skills cannot be accomplished well, which corresponds to a relatively low success rate. Even so, our method performs the best and manages to make more positive optimizations compared with stitching baselines.

The results related to `Door` $\rightarrow$ `CloseDoor` and `PickPlaceCan` $\rightarrow$ `PickPlaceCan` in Robosuite benchmark are listed in Table 3. The tasks in Robosuite are considered more complex and difficult than many other benchmarks (Xu et al., 2022; Dalal et al., 2024), and **MF-Stitching** makes positive optimizations over **No-Stitching**, while **Random-Stitching** yields negative effects. **MB-Stitching** (our method) reaches the best performance in both long-horizon tasks.

In addition, we provide further studies on hyperparameters of value ensemble on the Kitchen benchmark in Appendix C.2.

Table 3: Average scores of skill stitching on Robosuite. We evaluate our method as well as the baselines on stitching `Door → CloseDoor` and `PickPlaceCan → PickPlaceCan`, with each tested for 100 times for an average. The score is equal to the number of accomplished skills divided by the total number of skills (which is 2), i.e., the normalized score. We test on three different seeds and report the mean and variance.

| Tasks | Methods | | | |
|---|---|---|---|---|
| | **No-Stitch** | **Random-Stitch** | **MF-Stitch** | **MB-Stitch (Ours)** |
| Door → Close | 0.289±0.010 | 0.285±0.020 | 0.300±0.025 | **0.322±0.016** |
| Can → Can | 0.369±0.009 | 0.352±0.016 | 0.387±0.012 | **0.399±0.011** |

Table 4: Average scores of **Single-Policy** trained using the long-task dataset compared with **MB-Stitch**. The results show that the policy trained from long-horizon dataset can only accomplish the long tasks whose skills are in the same order with the dataset, exhibiting poor capabilities of generalization. We test on three different seeds and report the mean and variance.

| Tasks | Single-Policy | MB-Stitch | Tasks | Single-Policy | MB-Stitch |
|---|---|---|---|---|---|
| bot. | 0.000±0.000 | **0.943±0.031** | mic. → bot. | 0.565±0.011 | **0.840±0.006** |
| lig. | 0.000±0.000 | 0.000±0.000 | mic. → lig. | 0.485±0.004 | **0.705±0.011** |
| mic. | **0.917±0.025** | 0.883±0.021 | mic. → ket. | **0.914±0.011** | 0.742±0.003 |
| ket. | 0.013±0.009 | **0.327±0.025** | bot. → lig. | 0.072±0.009 | **0.554±0.017** |

| Tasks | Single-Policy | MB-Stitch |
|---|---|---|
| mic. → ket. → bot. | **0.682±0.006** | 0.563±0.002 |
| mic. → bot. → ket. | 0.374±0.007 | **0.564±0.008** |
| bot. → mic. → ket. | 0.062±0.004 | **0.382±0.003** |
| mic. → ket. → bot. → lig. | **0.544±0.001** | 0.424±0.003 |
| mic. → bot. → ket. → lig. | 0.289±0.001 | **0.440±0.003** |
| bot. → mic. → ket. → lig. | 0.053±0.002 | **0.366±0.006** |

## 4.3 ABLATION STUDY

### 4.3.1 USE TRAJECTORIES OF LONG-HORIZON TASKS FOR OFFLINE TRAINING

In this section, we explore the capability of the datasets of long-horizon tasks. The long-horizon dataset does not distinguish trajectories of each skill, but only containing the trajectories of the whole long task instead. We take this ablation study on the Kitchen benchmark, denoted as **Single-Policy**, with a long dataset that only contains the following long-horizon task: `microwave → kettle → bottom burner → light switch`. We train IQL policy with such a dataset and test the performances on various combinations of skills, with comparison to our method **MB-Stitch**. The results are listed in Table 4.

The results show that in terms of the tasks that contain skills with the same order exactly (`microwave`, `microwave → kettle`, `microwave → kettle → bottom burner`, and `microwave → kettle → bottom burner → light switch`), **Single-Policy** outperforms **MB-Stitch** (our method). However, for the other long-horizon tasks with different order of skills, **Single-Policy** cannot handle those out-of-distribution combinations and performs much worse than **MB-Stitching**.

It can be inferred that the policy trained from the long-horizon dataset can only accomplish the long-horizon tasks whose skills are in the same order with the dataset, indicating that **Single-Policy** exhibits poor capabilities of generalization in terms of different combinations of skills.

Table 5: Average scores of **No-Stitching** baseline and **MB-Stitching** on `Door` $\rightarrow$ `CloseDoor`, trained with 4 different levels of datasets respectively. From level-1 to level-4, the quality of the dataset gets higher. We test on three different seeds and report the mean and variance.

| Dataset | No-Stitching | MB-Stitching | Net Gain |
|---------|--------------|--------------|----------|
| level-1 | 0.292±0.009 | **0.314±0.021** | +7.55% |
| level-2 | 0.410±0.018 | **0.437±0.044** | +6.46% |
| level-3 | **0.584±0.004** | 0.537±0.007 | -8.05% |
| level-4 | 0.980±0.002 | 0.980±0.004 | 0% |

### 4.3.2 INFLUENCE OF DATASET QUALITY ON SKILL STITCHING

In this section, we explore the effect of stitching related to the quality of the dataset. We test the performances of skill stitching using different versions of the dataset, each with a different level of quality (from non-expert trajectories to expert ones). We adopt the dataset of `Door` and `CloseDoor` in the Robosuite benchmark with four different levels, in which level-1 denotes the non-expert (which is sub-optimal) dataset, and level-4 is the most expert one.

The results of different levels of the dataset are listed in Table 5. A dataset of lower level (representing the non-expert dataset) yields policies with worse performance, but brings higher positive effect of stitching. On the contrary, a dataset of higher level, which stands for a more expert dataset, yields policies with better performance, but with relatively lower effect of stitching and even negative optimization.

Analyzing this phenomenon, the more optimal the dataset we use, the better performance the policy will have, thus it can handle the next skill better even without the process of stitching. On the other hand, training with an expert dataset has more trend of "overfitting" - the policy can actually handle a smaller range of states since the variety of states in the expert dataset tends to be poor. Hence, the "overfitted" policy tends to result in a negative effect with the stitching process - the new state after stitching will probably become a state unseen in the expert dataset. In this aspect, policies trained from those non-expert datasets tend to be more capable in terms of "generalization".

## 5 RELATED WORK

**Online skill learning.**    This refers to the process of skill acquisition and skill stitching both taking place online. Konidaris & Barto (2009) was the first to introduce skill stitching as a method of skill discovery, such that an agent in an environment with continuous state space can construct a sequence of options to optimize the behaviors in the target MDP. However, it can only tackle simple environments with relatively low-dimensional states and discrete actions. Based on this, Deep Skill Chaining (DSC) (Bagaria & Konidaris, 2019) extends skill stitching to high-dimensional cases with continuous state and action spaces by leveraging hierarchical reinforcement learning. Inspired by the intuition that it is easier to solve a long-horizon task from the states in the local neighborhood of a goal (Bagaria & Konidaris, 2019), DSC defines the initial states of the current options as the terminal states for later options and learns the current options recursively. For instance, the initial state of the first option is placed near the goal, and the terminal state of the first option is the goal itself. Some other work, including Multi-Skill Mobile Manipulation (M3) (Gu et al., 2023), focused on the errors in skill stitching. For example, a stationary manipulation skill may perform poorly when applied in an inappropriate location. To overcome such issues, M3 incorporates navigation skills that are trained with region-based goals rather than point-specific ones, as well as manipulation skills that can offer enhanced mobility and flexibility.

**Offline skill acquisition with online stitching.**    Offline skills acquisition with online stitching refers to a framework where skills are pretrained or drawn from offline skill datasets, while the stitching policy is learned through online RL. In this context, the primary focus is on defining the initial and terminal state sets for these skills. Some work (Lee et al., 2019; Kang & Oh, 2022) specifically examined the design of the reward function for the stitching policy. In Lee et al. (2019),

the reward function incorporates a learned distance to the initial state of the next skill as a part of the reward signal. Similarly, in Kang & Oh (2022), the reward function is based on the learned distance to a certain cluster center of the next skill. To prevent the uncontrollable expansion of the initial state set, T-STAR (Lee et al., 2021) introduced Generative Adversarial Imitation Learning (GAIL) (Ho & Ermon, 2016) rewards, encouraging the agent to retain near-expert trajectories. Additionally, an adversarial framework is devised to regularize the terminal state distribution to ensure its remaining close to the initial set of the subsequent policy. Closely related to our work, Chen et al. (2023) proposed the Transition Feasibility Function, which measures the ability of the skills to succeed in the end when starting from a given state, and this function is learned using online reward signals. However, our research diverges by focusing on fully offline skill stitching, with no online learning or adaptation employed. Instead, we rely on the offline value function of the next skill to guide the process of stitching.

**Offline skill stitching.** To the best of our knowledge, there has been little work on fully offline skill stitching, where both skill learning and skill stitching are conducted offline. The most recent relevant work in this domain is STAP (Agia et al., 2023), which evaluates the feasibility of skills using Q-values learned from offline datasets, and generates action sequences via the cross entropy method (CEM). GSC (Mishra et al., 2023) introduces a generative framework for producing sequences of states and skill parameters using chained and skill-level diffusion models. Among these approaches, the skills are predefined with specific parameters, and CEM is used solely to search for the suitable skill parameters. By contrast, in our work, skills are represented as policies within MDPs, and we employ CEM directly to search in the action space for skill stitching. This distinction allows for a more flexible and direct integration of the skills in the offline setting.

## 6 CONCLUSION AND LIMITATIONS

In this paper, we explore the problem of offline skill stitching within the context of reinforcement learning. By formulating the problem and defining proper objectives, we propose a model-based approach that employs an ensemble of offline-trained dynamics models and a planning algorithm with conservative objectives for effective stitching execution. Our method outperforms various baselines, demonstrating improved performance across diverse combinations of long-horizon tasks in different benchmarks. These results highlight the promising potential of offline model-based algorithms in the field of skill stitching for addressing long-horizon tasks.

Our method has some limitations that could be addressed in future work. As discussed in Section 4.3.2, it can be challenging for our method to yield positive effects when the quality of the datasets is excessively high, meaning the skills are sufficient to handle long-horizon tasks without the need for stitching. Additionally, our approach may struggle to adapt to environments with partial observations. To address this issue, methods for training dynamics models in partial observable settings (Hafner et al., 2023) could be applied.

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

# A BENCHMARK DETAILS

In this section, we present brief introductions to the benchmarks we use in detail.

## A.1 MAZE RUNNER

Maze runner is $9 \times 9$ a maze-like environment for navigation, which is slightly modified from AMAGO (Grigsby et al., 2024). The initial location is always at the bottom center of the map. An agent navigates in the maze from the initial location with a discrete action space (up, down, left, right), and each action yields taking one step (grid) in the corresponding direction. The observation space is continuous, including the normalized absolute coordinates of both the current place and the location of the goal, as well as the normalized distance to the walls in the four directions. There are four fixed goals (denoted as $g_0, g_1, g_2, g_3$ in Figure 1) set in the maze environment in total, corresponding to the four skills to be accomplished.

Correspondingly, the four datasets contain the trajectories of accomplishing the four skills respectively. In the maze runner environment, a long-horizon task contains multiple ordered goals (e.g., $g_0 \rightarrow g_1 \rightarrow g_3 \rightarrow g_2$), and the agent ought to reach each goal in the same order by navigation. Arriving at a certain goal without reaching the previous goals will be invalid, and is not recorded as skill accomplishment.

## A.2 KITCHEN

Kitchen implements a 9-DoF Franka Panda robot placed in a simulated kitchen environment to manipulate several household items (Fu et al., 2020). There are seven skills in this environment, each corresponding to interacting with an item to reach a target state. By default, the robot always starts from a fixed posture. The 59-dimensional observation space of Kitchen includes the proprioception of the robot arm and the states of the seven objects, while the 9-dimensional action space is constructed with the control of the velocity of the joints and the gripper fingers. The seven skills (Fu et al., 2020) are defined as:

1. `Bottom burner`: twist the bottom control knob to activate the bottom burner of the stove.
2. `Top burner`: twist the top control knob to activate the top burner of the stove.
3. `Light switch`: twist the switch to turn on the light.
4. `Slide cabinet`: slide the door of the cabinet to the right to open it.
5. `Hinge cabinet`: open the door of a hinge cabinet by rotating it.
6. `Microwave`: open the door of the microwave.
7. `Kettle`: move the kettle from the bottom burner onto the top burner.

In this paper, we assume access to seven datasets, each containing the trajectories of a single skill. In the Kitchen benchmark, the long-horizon tasks are the combinations of several skills, e.g., `bottom burner` → `kettle` → `microwave` is a long-horizon task with three skills.

## A.3 ROBOSUITE

Robosuite (Zhu et al., 2020) is another simulation framework aimed at robot learning based on MuJoCo physics engine (Todorov et al., 2012). This benchmark provides a set of tasks, most of which focus on object manipulation. Both the action space and the observation space are continuous, whose dimensions vary from task to task. We take the following three skills into consideration:

1. `Door`: open a closed door until the angle between the door and the frame exceeds a threshold.
2. `Close door`: close an open door until the angle between the door and the frame is smaller than a threshold.
3. `PickPlaceCan`: pick up a can from the box on the one side, and place it into the box on the other side. The can is generated at the initial position with a randomization.

The tasks in Robosuite are considered more complex and difficult than many other benchmarks (Xu et al., 2022; Dalal et al., 2024), so we simplify the action spaces by adopting MAPLE (Nasiriany et al., 2022) as the low-level skills. MAPLE does not produce the accurate action in the same way as the original action space of Robosuite benchmark, but predicts the next pose of the robot and leverages motion control of the MuJoCo physics engine. The observation space includes both the proprioception of the robot arm and the states of the objects, whose dimension varies from skill to skill, depending on the object to be manipulated.

In this benchmark, we consider two kinds of long-horizon tasks:

- `Door` → `CloseDoor`. This refers to opening a closed door first, and close it afterwards.
- `PickPlaceCan` → `PickPlaceCan.` This refers to repeatedly pick up the can from one side to the other side twice. The instant when the first can is placed correctly, it disappears, and the second can appears (with randomization) at the initial location.

## B   DETAILS OF THE METHOD

### B.1   TRAINING POLICIES OF EACH SKILL AND DYNAMICS MODELS

We use implicit Q-learning (IQL) (Kostrikov et al., 2022) to train the policies of each skill. Specifically, we train a multi-task policy for the Maze Runner and Kitchen benchmark, while three different single-task policies are trained for `Door`, `Close door`, and `PickPlaceCan` in the Robosuite benchmark, respectively. For those trained with a multi-task policy, we concatenate each state with a one-hot vector representing the category of the current skill. We list the related hyperparameters for the IQL algorithm in Table 6.

Table 6: The hyperparameters used in the IQL algorithm.

| Hyperparameters | Value |
|---|---|
| learning rate | $3 \times 10^{-4}$ |
| dropout rate | 0.1 |
| learning rate decay | True |
| $\gamma$ | 0.99 |
| $\tau$ | 0.005 |
| expectile | 0.7 |
| temperature | 0.5 |
| batch size | 256 |

We use supervised learning to train the dynamics models, in which a two or three-layer MLP is employed with an activation function (`ReLU(·)` for Maze Runner, `ELU(·)` for Kitchen, and `Sigmoid(·)` for Robosuite).

### B.2   PERFORMING MODEL-BASED PLANNING

We utilize model predictive control (MPC) for model-based planning. To be concrete, we leverage random shooting method (Press, 2007; Bonnans, 2013) for Maze Runner, sampling the actions from a uniform distribution. We adopt a population size of 200, each containing actions of 5 steps, and predict the 200 new states after the 5 actions via the dynamics model. We choose the best individual by estimating the value of the new states using the value function of the IQL policy of the next skill, and execute this particular individual (5 steps of actions) as the stitching part.

For continuous environments including Kitchen and Robosuite, we adopt cross entropy method (CEM) (Rubinstein, 1999; De Boer et al., 2005), sampling the actions from a Gaussian distribution and optimize the mean and standard until convergence. Specifically, a population size of action sequences are sampled from the initial Gaussian distribution $\mathcal{N}(\mathbf{0}, \mathbf{1})$, and each sequence has a length of "horizon" (number of steps). The dynamics model predicts the new states after executing these

sampled individuals, and evaluate the values of the states via the value function of the IQL policy of the next skill. According to the estimated values, a number of elites are selected afterwards, whose actions are used to update the mean and standard of the Gaussian distribution. When achieving a max number of iterations of update, the CEM process terminates, and one step of action is finally executed in the environment according to the mean of the current Gaussian distribution. The hyperparameters used in CEM are listed in Table 7:

Table 7: The hyperparameters used in CEM planning.

| Hyperparameters | Kitchen | Robosuite |
|---|---|---|
| population size | 50 | 200 |
| number of elites | 10 | 30 |
| horizon (length of actions of each individual) | 3 | 3 |
| number of iterations | 300 | 200 |
| momentum (for soft update) | 0.9 | 0.9 |
| initial annealing temperature | 0.5 | 0.5 |
| discount factor of annealing temperature | 0.99 | 0.99 |
| $\alpha$ (used in value ensemble) | 7 | 10 |
| $\beta$ (used in value ensemble) | 5 | 10 (for `Door` / 20 (for `Can`) |

For the CEM method, only one step of action is really executed in the environment after a number of updating iterations. As long as the next state after this action does not trigger the termination condition, the process of CEM planning will take place again and form a loop. The termination condition is defined as a monotonicity of the (ensemble) value of the new state. When the new state does not meet the condition of monotonicity, i.e., the value of the new state is no higher than the last one, the process of the stitching part is terminated.

### B.3 OTHER TASK-RELEVANT HYPERPARAMETERS

In this section, we provide other hyperparameters related to different environments and tasks.

**The maximum steps for each environment.** In the Maze environment, the maximum steps for a task containing 2 / 3 / 4 skill(s) are all 50, including 5 steps in maximum of each stitching part between two adjacent skills. In the Kitchen environment, the maximum steps for 2 / 3 / 4 skill(s) are 250 / 360 / 480 respectively, including 20 steps in maximum of each stitching part between two adjacent skills. In the Robosuite environment, the maximum number of step for open door → close door is 50, and the maximum number of step for can → can is 100, including 20 steps in maximum of each stitching part between two adjacent skills.

## C ADDITIONAL RESULTS

### C.1 ADDITIONAL RESULTS ON KITCHEN BENCHMARK

We provide further results of the Kitchen benchmark in Table 8. Compared with Table 2, we calculate the average score over the skills except the last one. The results can better demonstrate that our method can handle a variety of long-horizon tasks with various skill combinations.

### C.2 TUNING HYPERPARAMETERS DURING VALUE ENSEMBLE

In this section, we study the influence of different combinations of hyperparameters of value ensemble on the performances of skill stitching. As shown in Table 9, we list different combinations of values of $\alpha$ and $\beta$ during value ensemble (Equation 2), as well as the corresponding experimental results comparing with **No-Stitching**. The results show that the performances of skill stitching is quite sensitive to the hyperparameters, and skill stitching does not always guarantee positive effects compared with **No-Stitching**.

Table 8: Additional results for kitchen benchmark. Compared with Table 2, we only calculate the average score over the skills before the last one. For example, X → Y → Z → kettle refers to an average over $A_6^3 = 120$ kinds of combinations. The results show that our method can outperform various baselines effectively. All the results are conducted on three different seeds, and we report the mean and variance.

| Tasks | Methods | | | |
|---|---|---|---|---|
| | **No-Stitch** | **Random-Stitch** | **MF-Stitch** | **MB-Stitch (Ours)** |
| X → bottom burner | 0.180±0.001 | 0.133±0.004 | 0.186±0.003 | **0.245±0.005** |
| X → top burner | 0.295±0.005 | 0.128±0.006 | 0.303±0.002 | **0.350±0.003** |
| X → light switch | 0.281±0.005 | 0.133±0.003 | 0.279±0.003 | **0.287±0.003** |
| X → slide cabinet | 0.310±0.001 | 0.152±0.004 | 0.319±0.001 | **0.326±0.007** |
| X → hinge cabinet | 0.183±0.002 | 0.127±0.002 | 0.189±0.003 | **0.221±0.003** |
| X → microwave | 0.116±0.002 | 0.093±0.005 | 0.130±0.001 | **0.164±0.002** |
| X → kettle | 0.203±0.000 | 0.094±0.003 | 0.184±0.000 | **0.219±0.002** |
| X → Y → bottom. | 0.114±0.000 | 0.085±0.001 | 0.113±0.001 | **0.148±0.001** |
| X → Y → top. | 0.191±0.001 | 0.087±0.001 | 0.182±0.001 | **0.199±0.001** |
| X → Y → light. | 0.189±0.000 | 0.106±0.002 | 0.191±0.002 | **0.215±0.002** |
| X → Y → slide. | 0.211±0.001 | 0.105±0.001 | 0.208±0.001 | **0.224±0.001** |
| X → Y → hinge. | 0.194±0.000 | 0.110±0.001 | 0.195±0.000 | **0.222±0.001** |
| X → Y → micro. | 0.103±0.001 | 0.074±0.001 | 0.117±0.001 | **0.133±0.001** |
| X → Y → ket. | 0.158±0.001 | 0.077±0.000 | 0.150±0.001 | **0.168±0.000** |
| X → Y → Z → bottom. | 0.086±0.000 | 0.068±0.000 | 0.089±0.000 | **0.113±0.000** |
| X → Y → Z → top. | 0.139±0.000 | 0.071±0.000 | 0.140±0.000 | **0.154±0.000** |
| X → Y → Z → light. | 0.142±0.000 | 0.085±0.000 | 0.148±0.000 | **0.170±0.000** |
| X → Y → Z → slide. | 0.143±0.000 | 0.082±0.000 | 0.145±0.000 | **0.165±0.000** |
| X → Y → Z → hinge. | 0.160±0.000 | 0.090±0.000 | 0.162±0.000 | **0.184±0.000** |
| X → Y → Z → micro. | 0.081±0.000 | 0.059±0.000 | 0.092±0.000 | **0.107±0.000** |
| X → Y → Z → ket. | 0.126±0.000 | 0.062±0.000 | 0.121±0.000 | **0.135±0.000** |

Value ensemble is adopted to avoid the over-estimation of the value of the state. As Equation 2 figures out, we alleviate the over-estimation with a minus of the variance to avoid the uncertainty during model-based planning. However, the most suitable values of hyperparameters can vary from skill to skill. Therefore, even the same combination of hyperparameters can result in different effects of skill stitching in terms of different long-horizon tasks.

Table 9: The influence of different combinations of hyperparameters of value ensemble on the performances of skill stitching. We list 9 combinations of different values of $\alpha$ and $\beta$ in the value ensemble (Equation 2) as well as the corresponding experimental results comparing with **No-Stitching**. The results show that the performances of skill stitching is quite sensitive to the hyperparameters, and skill stitching does not always guarantee positive effects compared with **No-Stitching**. We test on three different seeds and report the mean and variance.

| Methods | | Tasks | | |
|---|---|---|---|---|
| | | mic.$\rightarrow$bot. | ket.$\rightarrow$top. | mic.$\rightarrow$top.$\rightarrow$lig. |
| **No-Stitch** | | 0.805±0.014 | 0.222±0.024 | 0.466±0.011 |
| **MP-Stitch** | $\alpha = 1, \beta = 1$ | 0.375±0.011 | 0.358±0.028 | 0.293±0.004 |
| | $\alpha = 1, \beta = 10$ | 0.595±0.034 | 0.296±0.017 | 0.350±0.003 |
| | $\alpha = 1, \beta = 20$ | **0.839±0.011** | **0.531±0.002** | 0.463±0.004 |
| | $\alpha = 10, \beta = 1$ | 0.671±0.029 | 0.408±0.013 | 0.492±0.009 |
| | $\alpha = 10, \beta = 10$ | 0.825±0.021 | 0.435±0.025 | **0.576±0.007** |
| | $\alpha = 10, \beta = 20$ | 0.817±0.004 | 0.423±0.012 | 0.511±0.007 |
| | $\alpha = 20, \beta = 1$ | 0.610±0.009 | 0.454±0.017 | 0.426±0.004 |
| | $\alpha = 20, \beta = 10$ | 0.830±0.018 | 0.334±0.029 | 0.571±0.007 |
| | $\alpha = 20, \beta = 20$ | 0.768±0.015 | 0.371±0.013 | 0.519±0.007 |

