# OpenReview forum: "Offline Model-Based Skill Stitching"
_ICLR.cc/2025/Conference — ICLR 2025 Conference Withdrawn Submission_

### Official Review · Reviewer_gcaq · 2024-10-29

**Soundness:** 2
**Presentation:** 3
**Contribution:** 2
**Rating:** 5
**Confidence:** 3

**Summary:**

The paper introduces an algorithm for skill stitching from offline data,the algorithm has two phases, an offline training phase where each skill is extracted from an offline data that contains trajectories representing the skill. And a test phase, where the dynamics model is used for MPC-based skill stitching guided by the value function. The experiments demonstrate the performance of the method in comparison with some baselines; the ablations show that the quality of the data can have a significant effect on the performance of the skill changing as well as the diversity of transitions in the training distribution.

**Strengths:**

1. The introduction of the offline skill stitching problem is important for real-world applications

2. The idea of using a model and planning to stitch the skills is interesting and seems a good direction for further research.

3. The method results on the maze are strong compared to the baselines.

4. The results are better than baselines in general.

**Weaknesses:**

1. The assumption of the availability of a dataset for each skill is a strong assumption, is there a way to relax it? For example learning diverse skills from one offline dataset? Is this possible and is there any related work that focus on this problem?

2. Training each skill separately via offline RL seems expensive and time-consuming.

3. For some hyperparameters it is not clear to me they have been chosen, for example the maximum steps of skill execution seems very problem dependent.

4. The method does not seem effective on more complicated tasks (for example in table 2 the method fails in accomplishing more than one skill regardless of the number of skills in the task), but it is still better than the baselines.

**Questions:**

1. For the maze experiments, can you compare to offline goal conditioned RL for example goal-conditioned IQL?

2. For the MF-stitching baseline, do you train the model-free stitching policy for each two adjacent skills?

3. How does the method perform for each skills permutation? Is it better under some permutations and worse in others?

---

> ### Author Response · Authors · 2024-11-20
> **Thanks for your review! Here, we respond to your comments and address the issues. We hope to hear back from you if you have further questions!**
>
> ## Weaknesses:
>
> 1.	The assumption of the availability of a dataset for each skill is a strong assumption, is there a way to relax it? For example learning diverse skills from one offline dataset? Is this possible and is there any related work that focus on this problem?
>
> We are not sure what the "one dataset" here exactly refers to.
>
> - For one offline dataset with various single skills, we can divide it into parts of each skill.
> - For one offline dataset with only long-horizon tasks containing many skills in order, we can divide it into parts of each skill.
> - However, for one offline dataset with only one single skill, it might be unsolvable, since we have no access to the trajectories of other skills. The policies, value functions, and dynamics models trained on datasets of only one skill, are possibly unable to generalize to out-of-distribution states and transitions.
>
> 2.	Training each skill separately via offline RL seems expensive and time-consuming.
>
> We train a multi-task policy instead of multiple sing-task policies in both Maze and Kitchen, which can be viewed as a goal-conditioned policy. Take Kitchen as an example. Following the tradition of vector-based RL, we use a 7-dimensional one-hot vector (since there are 7 tasks in Kitchen) to denote which skill is being executed, and the one-hot vector is concatenated with the observation vector before taken by the policy network as input. We have made this clear in the revision (modified texts in blue in Section 4.1.2).
>
> 3.	For some hyperparameters it is not clear to me they have been chosen, for example the maximum steps of skill execution seems very problem dependent.
>
> The maximum steps for each environment are independent. To be concrete, in the Maze environment, the maximum steps for a task containing 2 / 3 / 4 skills are all 50, including 5 steps in maximum of each stitching part between two adjacent skills. In the Kitchen environment, the maximum steps for 2 / 3 / 4 skills are 250 / 360 / 480 respectively, including 20 steps in maximum of each stitching part between two adjacent skills. In the Robosuite environment, the maximum number of step for open door $\to$ close door is 50, and the maximum number of step for can $\to$ can is 100, including 20 steps in maximum of each stitching part between two adjacent skills. Note that we utilize action spaces referred from MAPLE, which greatly reduces the number of steps needed in the Robosuite benchmark compared to previous literature. We have made this clear in the revision (modified texts in blue in Appendix B.3).
>
> 4.	The method does not seem effective on more complicated tasks (for example in table 2 the method fails in accomplishing more than one skill regardless of the number of skills in the task), but it is still better than the baselines.
>
> This is highly due to the low success rate of the first skill, which is not caused by our stitching method. For example, the success rates of the 7 tasks in Kitchen are listed below:
>
> | Task | IQL Success Rate |
> |:------:|:------------------------:|
> | bottom burner | 0.96 |
> | top burner | 0.00 |
> | light switch | 0.00 |
> | slide cabinet | 0.00 |
> | hinge cabinet | 0.00 |
> | microwave | 0.88 |
> | kettle | 0.27 |
>
> Since our experimental results are based on the average over all possible permutations, the absolute value of scores will be inevitably low. For example, consider the first row of Table 2 in the paper, where we test over $A_7^2=42$ combinations of tasks. Among the final scores of these 42 tasks, at least $4\times 6=24$ of them will be 0, since 4 of the 7 tasks cannot be accomplished at all. Theoretically, the upper bound of the score accomplished by an arbitrary (even expert) algorithm should be $0.96 \times 2 \times \frac{1}{7} + 0.00 \times 2 \times \frac{4}{7} + 0.88 \times 2 \times \frac{1}{7} + 0.27 \times 2 \times \frac{1}{7} = 0.603$ (without normalize, as the revised version has normalized the results according to another reviewer). As shown in Table 2 (before revision), Our MB-Stitch has achieved 0.518, which is close to the upper bound, and outperforms the baselines.

---

> > ### Author Response · Authors · 2024-11-20
> > **Official Comment of Rebuttal (Cont'd)**
> >
> > ## Questions:
> >
> > 1.	For the maze experiments, can you compare to offline goal conditioned RL for example goal-conditioned IQL?
> >
> > In implementation, we already use one multi-task policy (i.e., the goal-conditioned policy) instead of four single-task policies to accomplish the four skills, since the goal information is naturally contained within the observations of the Maze environment. We have made this clear in the revision (modified texts in blue in Section 4.1.2).
> >
> > 2.	For the MF-stitching baseline, do you train the model-free stitching policy for each two adjacent skills?
> >
> > The model-free stitching policy in Maze or Kitchen is also a multi-task one, i.e., conditioned on a vector denoting which two adjacent skills are under consideration. The model-free stitching policies in Robosuite are two different ones (one is for door $\to$ close door; another is for can $\to$ can).
> >
> > 3.	How does the method perform for each skills permutation? Is it better under some permutations and worse in others?
> >
> > It is better under some permutations and worse in others. However, this is highly due to the magnitude of the gap during stitching. As shown in the table above (the table in the former rebuttal comment), the RL policies (IQL) are naturally suitable for some of the tasks, while relatively unsuitable for the others. Intuitively, a lower success rate of a particular skill indicates a lower suitability of executing this skill, and could results in a larger gap to stitch from the previous skill to this one. Empirically, a moderate gap can be suitable for our stitching method to handle. However, when the gap is too small, i.e., the RL policy itself can handle the next skill quite well without stitching, adding the stitching process will possibly lead to a negative effect, resulting in worse performances.

---

### Official Review · Reviewer_jg8h · 2024-11-04

**Soundness:** 2
**Presentation:** 2
**Contribution:** 2
**Rating:** 3
**Confidence:** 4

**Summary:**

This work explores a model-based approach for offline learning of skills and their sequential stitching using only individual skill datasets, without relying on online interactions with the environment. Unlike existing skill stitching techniques based on online reinforcement learning, this approach utilizes offline data to decompose long-horizon tasks into manageable skills that can be executed sequentially. The focus is on training a dynamics model with aggregated skill datasets, enabling effective model-based planning and incorporating conservative optimization objectives to ensure robust transitions between skills during planning.

**Strengths:**

- The proposed offline skill stitching method is straightforward yet effective in certain environments with long-horizon tasks, enabling task completion by sequencing learned skills from offline datasets.
- Skill stitching offers a practical approach in hierarchical reinforcement learning, addressing challenges in learning tasks composed of multiple sub-tasks.

**Weaknesses:**

- Lack of novelty: The proposed skill stitching method of evaluating states for stitching using the value function is not novel; it is a fundamental approach used in existing offline RL for trajectory stitching [1, 2]. A comparison with these existing offline trajectory stitching methods is required.

[1] Stitching Sub-trajectories with Conditional Diffusion Model for Goal-
Conditioned Offine RL (AAAI 2024)

[2] Model-based Trajectory Stitching for Improved Offline Reinforcement
Learning (NeurIPS 2023)

The below work also uses model-based rollouts (planning) for skill-based task planning in offline settings, similar to the proposed method.

[3] Offline Policy Learning via Skill-step Abstraction for Long-horizon Goal-Conditioned Tasks (IJCAI 2024)

-	The proposed method using MPC operates by sampling possible actions and evaluating the value of the resulting states. For continuous action spaces, it requires extensive sampling and evaluation to determine the best outcome. Furthermore, in environments with stochasticity, the MPC optimization can be required at each attempt, leading to significant inefficiencies in time complexity.

- The performance gain in the Kitchen appears minimal, raising questions about whether the proposed method is effective in continuous action space settings. In the Maze Runner, the discrete action space makes the MPC method feasible. However, in complex continuous tasks like the Kitchen task, the value function evaluation may be unreliable, requiring MPC to extensively search the possible action space, which may explain the minimal performance gain observed.

- The method may not generalize well across diverse environments, especially those with dynamic or unpredictable conditions, as it relies solely on offline data without any consideration on real-time adaptability.

- The approach's effectiveness is highly dependent on the diversity of the offline datasets, as the method relies on the learned dynamics model on the aggregated offline datasets.

**Questions:**

- I wonder if the value function properly evaluates states that have not been visited (during stitching). As the value functions for each skill are learned distinctly, how can the value evaluation in the stitched space be accurate and reliable?
- How might the proposed method be adapted to handle low-coverage offline datasets?
- I wonder if the authors considered any techniques to reduce the computational burden of MPC in continuous or stochastic environment?
- What potential strategies could be considered for improving generalization or adaptability to dynamic environments within the constraints of offline learning?

- Minor Typos:

line 97: over-estimate → overestimate, to match the usage elsewhere in the paper.

line 215: continous actions space → continuous action space

line 257: T(\cdot|s_t,a_t) → T_{\phi}(\codt|s_t,a_t}

---

> ### Author Response · Authors · 2024-11-20
> **Thanks for your review! Here, we respond to your comments and address the issues. We hope to hear back from you if you have further questions!**
>
> ## Weaknesses:
>
> 1. Lack of novelty:
>
> [1] trains a critic $Q(s, a)$ with goal-relabeled short sub-trajectories, then trains a value-conditioned diffusion model to generate trajectories given a goal, and finally executes the actions in the generated trajectories. [2] introduces trajectory stitching for synthesizing new trajectories (as a method of data augmentation), and trains BC policies over augmented datasets. Although [3] uses model-based rollouts (planning) for skill-based task planning, it has less to do with skill stitching.
>
> Unlike previous literature [1, 2], our paper lies in a totally different problem setting, where we regard each task as a skill. [1] and [2] both stitch sub-trajectories within a single task, while we consider task-level stitching in our paper. When generating an extra sub-trajectory $\tau_2$ to be stitched after the former sub-trajectory $\tau_1$, there is no gap between $\tau_1$ and $\tau_2$. However, when stitching two task-level skills (e.g., microwave and kettle), the gap between two adjacent tasks is rather large, which is the biggest difference in between. In a word, our method tackles the challenge of skill stitching in the case where there are large gaps between two task-level skills, which is different from previous works [1, 2].
>
> [1] Stitching Sub-trajectories with Conditional Diffusion Model for Goal- Conditioned Offline RL (AAAI 2024)
>
> [2] Model-based Trajectory Stitching for Improved Offline Reinforcement Learning (Offline RL Workshop @ NeurIPS 2022)
>
> [3] Offline Policy Learning via Skill-step Abstraction for Long-horizon Goal-Conditioned Tasks (IJCAI 2024)
>
> ## Questions:
>
> 1.	I wonder if the value function properly evaluates states that have not been visited (during stitching). As the value functions for each skill are learned distinctly, how can the value evaluation in the stitched space be accurate and reliable?
>
> Learning the value functions distinctly for each skill does not mean the inaccuracy. As long as the used value function represents the relative suitability (possibility) of accomplishing the next skill (task), it can be utilized to guide the process of stitching. Besides, we adopt the trick of model ensemble to reduce the inaccuracy of value functions, which can make value estimation more reliable.
>
> 2.	How might the proposed method be adapted to handle low-coverage offline datasets?
>
> In the phase of learning individual skills, our method faces the same challenges as offline RL methods when data coverage is limited. However, in the skill stitching phase, our approach can mitigate the scarcity of each individual skill dataset by training on the union of all datasets, where the necessary transitions between skills are more likely to be present. Additionally, our model-based approach leverages the generalization capabilities of world models to fill in missing transitions, while incorporating a conservative strategy to mitigate distribution shift.
>
>
> 3.	I wonder if the authors considered any techniques to reduce the computational burden of MPC in continuous or stochastic environment?
>
> To accelerate MPC, techniques such as approximate planning (e.g., TD-MPC [4], which uses learned value functions to bootstrap planning), batch computation, model compression and distillation, and receding horizon control [5] could be applied. While these methods are promising, we have not focused on efficiency in this work, as our primary contribution is improving the stitching performance of offline-learned skills.
>
> [4] Hansen et al., "Temporal difference learning for model predictive control." (2022).
>
> [5] Mayne and Michalska. "Receding horizon control of nonlinear systems." (1988)
>
> 4.	What potential strategies could be considered for improving generalization or adaptability to dynamic environments within the constraints of offline learning?
>
> Data augmentation could be properly considered, utilizing generative models. Through data augmentation, we could get larger quantities of data with a higher coverage of states and transitions.
>
> 5.	Minor typos.
>
> Thank you for pointing out the typos! We have corrected these in red color in the revision.

---

> > ### Comment · Reviewer_jg8h · 2024-11-28
> >
> > I appreciate your diligent response. While I have carefully reviewed the distinctions you highlighted between your work and existing research on skill stitching, my concerns regarding the originality of this paper remain only partially addressed. Certainly, tackling long-horizon tasks through task-level stitching is a significant challenge, and I find the approach presented in your paper practical. However, the detailed methodology, particularly how the model is employed to address gaps in different sub-trajectories, does not appear to be technically novel. For instance, the multi-step goal chaining described in Section 3.2 of [1] and the model-guided rollouts illustrated in Figure 3 of [3] share similarities in both their approach and purpose with the model-based planning proposed in this paper. I think it would have strengthened the work if these prior studies were compared more thoroughly and included as baselines in the experiments. While a certain degree of technical similarity does not necessarily diminish the value of a paper, my concern lies in the heavy reliance on model-based planning for much of the contribution. As such, I will maintain my original score.

---

### Official Review · Reviewer_sWqG · 2024-11-04

**Soundness:** 2
**Presentation:** 2
**Contribution:** 2
**Rating:** 3
**Confidence:** 4

**Summary:**

This paper investigates the development of agents capable of addressing long-horizon tasks through offline model-based reinforcement learning (RL). While current RL methods excel at learning individual skills, they struggle with integrating these skills to accomplish extended tasks due to the mismatch between the termination of one skill and the initiation of another, resulting in distribution shifts. The authors propose an offline approach to skill stitching, leveraging aggregated datasets from various skills to train a dynamics model that can generalize across different skills. This model, along with an ensemble of offline dynamics models and value functions, is used to stitch adjacent skills through model predictive control (MPC). To address the overestimation issues common in offline model learning, a conservative method is introduced to penalize uncertainty in model and value predictions. The study's experimental results demonstrate the effectiveness of this approach over baseline methods in offline settings across multiple benchmarks.

**Strengths:**

1. This paper is written well. The method is esay to follow.
2. This work is evaluted on various domains.

**Weaknesses:**

1. The originality of this work is quietly limited. The idea of stitching skills based on value functions is not new; many papers have proposed similar approaches. For example, PEX [1].
2. A large number of baseline algorithms are missing. For example, OPAL [2] and LPD [3].


[1] Zhang, Haichao, We Xu, and Haonan Yu. "Policy expansion for bridging offline-to-online reinforcement learning." arXiv preprint arXiv:2302.00935 (2023).

[2] Ajay, A., Kumar, A., Agrawal, P., Levine, S., & Nachum, O. (2020). Opal: Offline primitive discovery for accelerating offline reinforcement learning. arXiv preprint arXiv:2010.13611.

[3] Yang, Y., Hu, H., Li, W., Li, S., Yang, J., Zhao, Q., & Zhang, C. (2023, June). Flow to control: Offline reinforcement learning with lossless primitive discovery. In Proceedings of the AAAI Conference on Artificial Intelligence (Vol. 37, No. 9, pp. 10843-10851).

**Questions:**

1. What if the model learns inaccurately in a complex environment?

2. Can you use the normalized score for the experimental results?

---

> ### Author Response · Authors · 2024-11-20
> **Thanks for your review! Here, we respond to your comments and address the issues. We hope to hear back from you if you have further questions!**
>
> ## Weaknesses:
>
> 1. About originality and baselines:
>
> PEX [1], OPAL [2] and LPD [3] all tackle challenges in single long-horizon tasks. For example, PEX [1] introduces offline-trained policy $ \pi_\beta $ and online-trained policy $ \pi_\theta $, but these two policies are used as a policy pool to execute within a single task. During the execution phase, an action is produced from the joint policy $\Pi=[\pi_\beta, \pi_\theta]$ for each step. OPAL [2] introduces hierarchical structures to stitch primitive policies.
>
> Different from previous literature, our paper lies in a totally different problem setting, where we regard each task as a skill. In other words, we consider task-level stitching. The largest gap between primitive-level stitching and task-level stitching is that task-level stitching requires inserting an unseen trajectory sequence, which is not needed in primitive-level stitching. Considering an example of task-level stitching: microwave $\to$ kettle, there is a giant gap between the last state of finishing the task “microwave” and the initial state to accomplish the task “kettle”. However, for a primitive-level case such as “kettle = move + grasp + lift”, there is no gap when switching from “grasp” to “lift”. In a word, our method tackles the challenge of skill stitching in the case where there are large gaps between two task-level skills, which is different from previous works [1, 2, 3].
>
> [1] Zhang, Haichao, We Xu, and Haonan Yu. "Policy expansion for bridging offline-to-online reinforcement learning." arXiv preprint arXiv:2302.00935 (2023).
>
> [2] Ajay, A., Kumar, A., Agrawal, P., Levine, S., \& Nachum, O. (2020). Opal: Offline primitive discovery for accelerating offline reinforcement learning. arXiv preprint arXiv:2010.13611.
>
> [3] Yang, Y., Hu, H., Li, W., Li, S., Yang, J., Zhao, Q., \& Zhang, C. (2023, June). Flow to control: Offline reinforcement learning with lossless primitive discovery. In Proceedings of the AAAI Conference on Artificial Intelligence (Vol. 37, No. 9, pp. 10843-10851).
>
> ## Questions:
>
> 1.	What if the model learns inaccurately in a complex environment?
>
> The inaccuracy of value functions and dynamics models will potentially lead the stitching process to a sub-optimal state, thus resulting in a failure during task execution. In our work, we adopt model ensemble to reduce the inaccuracy of both value functions and dynamics models.
>
> 2.	Can you use the normalized score for the experimental results?
>
> We use the absolute scores (the number of tasks accomplished) previously since we study the problem of task-level skill stitching. Normalized scores are just $\frac{\text{absolute scores}}{\text{the number of tasks in total}} \times 100\\%$. We have modified all the experimental results into normalized scores, as shown in the tables with green color in the revised paper.

---

> > ### Comment · Reviewer_sWqG · 2024-11-25
> >
> > Thanks for your replay. I still be concerned about the experiments. The gap between primitive-level stitching and task-level stitching may be large. However, it is still valuable to verify the primitive-level stitching methods in these tasks. Therefore, I keep my score.
> >
> > Suggestions: The experimental results in Table 1-4 are hard to follow. You can find ways to make them more understandable.

---

### Note · Authors · 2024-12-05

I have read and agree with the venue's withdrawal policy on behalf of myself and my co-authors.